# RNA Binding Protein PTBP1 Promotes the Metastasis of Gastric Cancer by Stabilizing *PGK1* mRNA

**DOI:** 10.3390/cells13020140

**Published:** 2024-01-12

**Authors:** Xiaolin Wang, Ce Liang, Shimin Wang, Qiang Ma, Xiaojuan Pan, Ai Ran, Changhong Qin, Bo Huang, Feifei Yang, Yuying Liu, Yuying Zhang, Junwu Ren, Hao Ning, Haiping Li, Yan Jiang, Bin Xiao

**Affiliations:** 1College of Pharmacy, Chongqing Medical University, Chongqing 400016, China; 2021320057@stu.cqmu.edu.cn (X.W.); yehanlianjiangyubl@163.com (C.L.); 18380290874@163.com (S.W.); maq705@163.com (Q.M.); pxj07071996@163.com (X.P.); 15123449792@163.com (A.R.); 15928374227@163.com (C.Q.); zmc.hb@foxmail.com (B.H.); ffyang18@163.com (F.Y.); 18090279222@163.com (Y.L.); yuyingzhang9901@163.com (Y.Z.); 13637703800@163.com (J.R.); hning@stu.cqmu.edu.cn (H.N.); lhaiping1994@163.com (H.L.); jerryyanki@163.com (Y.J.); 2Key Laboratory of Basic Pharmacology of Ministry of Education and Joint International Research Laboratory of Ethnomedicine of Ministry of Education, Zunyi Medical University, Zunyi 563006, China

**Keywords:** PTBP1, gastric cancer, PGK1, metastasis

## Abstract

Gastric cancer (GC) is the most common type of malignant tumor within the gastrointestinal tract, and GC metastasis is associated with poor prognosis. Polypyrimidine tract binding protein 1 (PTBP1) is an RNA-binding protein implicated in various types of tumor development and metastasis. However, the role of PTBP1 in GC metastasis remains elusive. In this study, we verified that PTBP1 was upregulated in GC tissues and cell lines, and higher PTBP1 level was associated with poorer prognosis. It was shown that PTBP1 knockdown in vitro inhibited GC cell migration, whereas PTBP1 overexpression promoted the migration of GC cells. In vivo, the knockdown of PTBP1 notably reduced both the size and occurrence of metastatic nodules in a nude mice liver metastasis model. We identified phosphoglycerate kinase 1 (PGK1) as a downstream target of PTBP1 and found that PTBP1 increased the stability of *PGK1* by directly binding to its mRNA. Furthermore, the PGK1/SNAIL axis could be required for PTBP1’s function in the promotion of GC cell migration. These discoveries suggest that PTBP1 could be a promising therapeutic target for GC.

## 1. Introduction

Gastric cancer (GC) is the most prevalent digestive tumor globally and ranks as the fourth leading cause of cancer-related deaths [1]. Despite advancements in understanding the biology of GC, surgical or endoscopic resection remains the primary treatment approach [2]. GC cells are able to invade lymph, peritoneum, blood, and neighboring organs. And liver metastasis, peritoneal metastasis, lung metastasis, bone metastasis, and lymph node metastasis are common in GC. The metastasis process of GC is a dynamic multi-step biological cascade, regulated by complex factors. The metastasis process can be roughly divided into the following steps [3]: (1) GC cells shed from the primary focus, invade the basement membrane, and then invade the blood vessels, lymphatic vessels, and peritoneum; (2) a few GC cells survive in the circulatory system and then spread to other tissues and organs along with the circulatory system; (3) GC cells form micro-metastases in the appropriate “soil” and continue to proliferate to form secondary tumors of the same type as the primary tumor, which can continue to invade and metastasize. 

RNA binding proteins are a class of highly conserved proteins that exist in the nucleus and cytoplasm and participate in post-transcriptional regulation in all manner of RNA biology [4]. More than 2000 RNA binding proteins (RBPs) have been identified in recent years. It is validated that RBPs can interact with various types of RNAs (including mRNAs, ncRNAs, tRNAs, snRNAs, and snoRNAs), and mounting evidence suggests that dysregulated RBPs are connected with cancer metastasis. PTBP1 is a known RBP that can control messenger RNA (mRNA) splicing, translation, stability, and localization [5]. Recent studies show that PTBP1 plays an important role in the occurrence and development of cancer, through regulating cell proliferation, glycolysis, and apoptosis, as well as angiogenesis [6].

In this study, we demonstrated that PTBP1 was significantly highly expressed in GC and was strongly associated with the poor prognosis of GC patients. PTBP1 significantly promoted the migration of GC cells in vitro and in vivo. Mechanistically, PTBP1 increased the expression of PGK1 by directly binding to *PGK1* mRNA and enhancing the stability of *PGK1* mRNA. The RNA recognition motifs RRM1 and RRM3 of PTBP1 were involved in the interaction between PTBP1 and *PGK1* mRNA. PTBP1-PGK1-SNAIL regulation axis may play an important role in the metastasis of GC. 

## 2. Materials and Methods

### 2.1. Clinical Specimens

Four pairs of gastric cancer tissues and corresponding noncancerous tissues were obtained from Southwest Hospital, Army Military Medical University, China. All cases were confirmed by histopathological examination and were without chemotherapy or radiotherapy before surgery. Resected specimens were collected immediately into RNA-later (Thermo Scientific, Pittsburgh, PA, USA) and stored at −80 °C. The study was approved by the Ethics Committees of Chongqing Medical University and Southwest Hospital of Army Military Medical University. Written informed consent was also obtained from all patients.

### 2.2. Cell Culture

Normal gastric epithelial cell GES-1 and GC cell lines SGC7901 and HGC27 were obtained from Army Medical University (Chongqing, China). MKN-74 and AGS cells were purchased from Meisen (Hangzhou, China). All cells were cultured in DMEM, F12, or RPMI 1640 medium (basal medium, Shanghai, China) supplemented with 10% fetal bovine serum (Lonsera, Montevideo, Uruguay) at 37 °C with 5% CO_2_.

### 2.3. Bioinformatic Analysis

The expression levels of PTBP1 and PGK1 in cancers were obtained by analyzing the TCGA database, TIMER2.0 (http://timer.cistrome.org/; accessed on 13 August 2022), and Gene Expression Profiling Interactive Analysis (GEPIA) (http://gepia2.cancer-pku.cn/#index; accessed on 27 September 2022). The survival curves of PTBP1 and PGK1 in GC patients were from the Kaplan–Meier Plotter (https://kmplot.com/analysis/; accessed on 12 December 2022). Seven genes (*Snail*, *Slug*, *MMP2*, *MMP3*, *MMP9*, *Twist*, and *Zeb1*) were thought to be linked to epithelial–mesenchymal transition (EMT). We put these 7 genes into GEPIA as signatures and calculated the correlation with PTBP1 and PGK1. FunRich (http://www.funrich.org; accessed on 15 February 2023) allows for complete database customization and thereby permits the tool to be exploited as a skeleton for enrichment analysis. We put 184 genes from the overlap of si-PTBP1 RNA-Seq data and the GEPIA data in FunRich to further investigate the biological functions of the genes.

### 2.4. Cell Transfections and Lentiviral Transduction

The siRNAs were designed and produced by Gene Pharma (Appendix A). Briefly, cells seeded overnight in 6-well plates were transfected with siRNA using Liposome 2000 (Invitrogen, Carlsbad, CA, USA) according to the manufacturer’s instructions and incubated for 6 h before replacement with a serum-containing medium. Transient overexpression of PTBP1 was performed by inserting the CDS region of PTBP1 into the pcDNA3.1 vector. Full-length and truncated PTBP1 plasmids were constructed in the pCDH vector. Neofect (Neofect, Beijing, China) was used to transfect plasmids into GC cells. To construct cell lines with a stable knockdown of PTBP1, we used the HBLV-LUC-PURO vector and synthesized the lentivirus at HanBio (Shanghai, China). The lentivirus sequence was identical to that of siRNA (Appendix A). HGC27 cells (Chongqing, China) were seeded equally into 24-well plates and placed in an incubator overnight. Lentiviral vectors encoding small interfering RNA targeting PTBP1 and control lentivirus CON207 were then transfected. Next, they were treated with puromycin (2 μg/mL) for 1 week. The transfection efficiency of the virus was verified via Western blot, and these cells were used for subsequent experiments.

### 2.5. Quantitative Real-Time PCR

Total RNA was extracted from RNAiso Plus (Takara, Kyoto, Japan) and reverse transcribed with the PrimeScript RT–PCR kit (Takara, Kyoto, Japan) per the manufacturer’s instructions. Quantitative real-time PCR assays were finalized with SYBR Green Master Mix (Bioground, Chongqing, China) with CFX connect Real-Time PCR System (BioRad, Hercules, CA, USA) at the recommended thermal cycling settings: an initial cycle at 95 °C for 15 min, followed by 40 cycles of 15 s at 95 °C and 30 s at 61.9 °C (PTBP1), 58 °C (β-ACTIN, PGK1). Relative mRNA expression was calculated in the 2^(−ΔΔCt)^ method. Appendix A contains information on the primer sequences.

### 2.6. Western Blot

Total proteins were extracted from GC tissues and cells using RIPA lysis buffer (Beyotime, Shanghai, China) and denatured in a 100 °C metal bath for 10 min. Based on their sizes, proteins were separated by SDS-PAGE and transferred onto PVDF membranes. The transferred proteins were blocked using 5% skimmed milk for 1 h and incubated overnight at 4 °C with an appropriate dilution of the primary antibody. The antibodies used in this study were as follows: PTBP1 (# 32-4800, Invitrogen, Carlsbad, CA, USA), PGK1 (# ST49-07, HUABIO, Zhejiang, China), FLAG (# 14793, CST, USA), VINCULIN (# JM42-43, ZENBIO, Chengdu, China), SNAIL (# C15D3, CST, Danvers, MA, USA), β-ACTIN (# AA128, Beyotime, Shanghai, China), and Epithelial–Mesenchymal Transition (EMT) Antibody Sampler Kit (#9782, CST, Danvers, MA, USA).

### 2.7. Transwell Assay

For the transwell assay, 7 × 10^4^ HGC27 cells or 5 × 10^4^ AGS cells (without fetal bovine serum) were digested from 6-well plates and then inoculated into the upper chamber of 8 μm chambers (Corning, Corning, NY, USA). Then, 500 μL of medium containing 10% FBS was added to the lower chamber. After 24 h of incubation, the non-migrating cells in the upper chamber were gently wiped, and the migrating cells at the bottom of the chamber were fixed with 4% paraformaldehyde for 10 min, followed by staining with the crystalline violet solution for 8 min. The pictures were obtained and the cells were counted under a microscope.

### 2.8. Wound Healing

In the wound healing experiment, first, cells were grown in 6-well plates and then the monolayer was gently scraped off using a cell scratcher when the cell confluency reached approximately 80%. Subsequently, the wells were washed twice with phosphate-buffered saline (PBS), and a fresh medium containing 1% serum was added. Images were acquired at 0 h, 12 h, and 24 h using an optical microscope.

### 2.9. Animal Experiments

We adopted a novel modified mouse model of liver metastasis in order to measure the formation of GC liver metastases [7,8]. Six-week-old male BALB/c nude mice were used in this study. In order to establish mouse models of liver metastasis, gastric cancer cells (sh-NC-HGC27 and sh-PTBP1-HGC27) were mixed in 100 µL serum-free medium at a density of 1 × 10^6^ cells/animal. Then, the spleens were removed to prevent the colonization of GC cells in the spleen. The mice were sacrificed after 50 days. Mice’s livers were removed carefully and preserved in 4% formalin. The liver tissues were stained with hematoxylin and eosin (HE) in order to identify any instances of liver metastasis. The Chongqing Medical University Animal Management Committee approved all of the study’s animal trials.

### 2.10. RNA Binding Protein Immunoprecipitation Assay

RIP experiments were accomplished with the Magna RIP Kit (Millipore, Merk Millipore, Darmstadt, Germany). The SGC-7901 cells were lysed by adding a lysis buffer containing protease and RNase inhibitors for 5 min. Inhibitor-containing lysate was added at a rate of 125 μL per 1 × 10^7^ cells/reaction density. The magnetic beads were conjugated to 8 μg antibody against PTBP1 (Merk Millipore, Darmstadt, Germany), Flag (CST, Danvers, MA, USA), and control IgG at indoor temperature, and the corresponding cell lysates were added. Then, the samples were incubated overnight at 4 °C, followed by RNA purification and protein extraction. Eventually, the RIP efficiency was verified via Western blot assay; the enrichment of PGK1 was verified via RT-qPCR assay.

### 2.11. RNA Sequencing

To investigate the downstream targets of PTBP1, RNA sequencing was performed in AGS cells after PTBP1 silencing. AGS cells were seeded in 6-well plates at a density of 4 × 10^5^ cells per well. After 24 h, siRNA targeting PTBP1 was transfected using liposome 2000 reagent (Invitrogen) recommended by the manufacturer. After 6 h, the supernatant was replaced with fresh medium containing 10% FBS. Cells were harvested after 48 h. At the same time, to explore the RNA that can bind to PTBP1 protein, the RNA that can bind to endogenous PTBP1 protein in SGC-7901 cells was pulled down and sequenced by RIP assay. One μg of RNA was used as input material for RNA sample preparation. In brief, mRNA molecules containing poly-A were refined using poly-T oligomeric magnetic beads. Subsequently, mRNA was cleaved into small fragments containing divalent cations for 10 min at 94 °C. The cleaved RNA fragments were then transcribed into first-strand cDNA using reverse transcriptase and random primers. This was followed by second-strand cDNA synthesis using DNA Polymerase I and RNase H. Then, cDNA fragments went through an end repair process, which involved the addition of a single ‘A’ base, and the ligation of the adapters. Construction of the final cDNA library was completed by purification and PCR enrichment. Library construction and sequencing were performed by Zhongke Genomics, Shanghai, China.

### 2.12. Actinomycin D Assay

AGS and HGC27 cells with PTBP1 knockdown were cultured in 12-well plates and then treated with actinomycin D (5 μg/mL) (Genview, Beijing, China) or corresponding DMSO for the designated durations. The cells were lysed in RNAiso Plus (Takara, Kyoto, Japan) for RNA extraction, and the mRNA stability was verified via RT-qPCR measurements.

### 2.13. Statistical Analysis

Statistical analysis was performed with GraphPad Prism 8 (GraphPad Software, San Diego, CA, USA). Student’s *t* tests were used to assess the differences in continuous variables between the two groups. *p* values < 0.05 was defined as statistically significant.

## 3. Results

### 3.1. PTBP1 Is Highly Expressed and Is a Poor Prognostic Factor in GC

We first analyzed the expression of PTBP1 in GC using TIMER 2.0 and GEPIA (http://gepia2.cancer-pku.cn/#index, Peking University, Beijing, China). The results showed that *PTBP1* mRNA expression was significantly increased in GC tissues compared to adjacent tissues (Figure 1A,B). Subsequently, we confirmed the upregulation of PTBP1 protein in GC tumor tissues via Western blot analysis (Figure 1C). Furthermore, PTBP1 expression was upregulated in all four GC cell lines (SGC7901, AGS, MKN74, and HGC27) compared to the gastric mucosal epithelial cell line (GES-1), which was confirmed by RT-qPCR and Western blot (Figure 1D,E). Then, the Kaplan–Meier Plotter database evidenced that GC patients with higher PTBP1 expression exhibited considerably shorter times of overall survival (OS), first progression (FP), and post-progression survival (PPS) than that of the patients with lower expression (Figure 1F–H). In addition, seven genes linked to epithelial–mesenchymal transition (EMT) (*Snail*, *Slug*, *MMP2*, *MMP3*, *MMP9*, *Twist*, and *Zeb1*) were inserted into GEPIA as a signature to gauge their correlation with PTBP1. The analysis revealed a correlation between PTBP1 and EMT-associated genes (Figure 1I). Subsequently, we examined the change in EMT-related proteins, including ZO-1, E-Cadherin, Claudin-1, and Slug in AGS and HGC27 cells after PTBP1 interference. We found that si-PTBP1 resulted in the upregulation of ZO-1 and E-Cadherin protein levels, which is favorable for the activation of EMT; however, it had no significant effect on the expression of Claudin-1 and Slug (Figure 1J). These results suggest that PTBP1 is upregulated in GC. It is a poor prognostic factor and may be involved in the EMT process of GC.

### 3.2. PTBP1 Promotes GC Metastasis In Vitro and In Vivo

To test whether PTBP1 affects GC cell metastasis, PTBP1 was knocked down using siRNA (si-PTBP1) or overexpressed using GV141-PTBP1 plasmid in AGS and HGC27 cells. The efficiency of silencing and the overexpression of PTBP1 was then determined by RT-qPCR and Western blot (Figure 2A,B). Transwell assay and wound-healing assay showed that silencing PTBP1 expression significantly reduced and inhibited the number and rate of GC cell migration. Meanwhile, the overexpression of PTBP1 significantly promoted GC cell migration (Figure 2C,D). To evaluate the effect of PTBP1 on the metastatic ability of gastric cancer cells in vivo, we established a liver metastasis model of gastric cancer in nude mice. First, we established a stable sh-PTBP1-HGC27 cell line using lentivirus (Figure 2E). The sh-PTBP1-HGC27 cells were then injected into nude mice’s spleens, and then the spleens were removed to prevent the colonization of GC cells in the spleen. Those nude mice were sacrificed on the 50th day. (Figure 2F). We observed a reduced number and size of hepatic metastatic nodules on the surfaces of the liver in the sh-PTBP1 group compared with the control group. (Figure 2G). Furthermore, HE staining in sh-PTBP1 group confirmed the presence of hepatic metastasis with typical pathologic characteristics, including neoplastic cells, high nuclear-cytoplasmic ratio cells, fibrosis, and necrosis (Figure 2H). The above results suggest that PTBP1 promotes GC metastasis in vitro and in vivo.

### 3.3. PTBP1 Positively Regulates the Expression of PGK1 in GC

To elucidate the molecular mechanisms regulating gastric cancer metastasis, we investigated the downstream target genes of PTBP1. First, high-throughput transcriptome sequencing was performed on PTBP1-knockdown AGS cells, and a total of 865 differentially expressed genes (including 361 downregulated and 504 upregulated genes) were identified (si-PTBP1 RNA-Seq) under the screening condition of Q-value < 0.05, which required FPKM > 1 in all groups (Appendix A). Meanwhile, 4640 differentially expressed genes were obtained from the GEPIA gastric cancer database (Appendix A). We obtained 184 genes (Figure 3A). Next, 184 genes were identified through the overlap of si-PTBP1 RNA-Seq data with the GEPIA data. To further investigate the biological functions of the overlapping genes, we performed a biological pathway analysis of those 184 genes in FunRich [9]. As a result, 35 genes were revealed to be associated with mTOR and AKT- PI3K signaling pathways (Figure 3B). 

PTBP1 is an RNA-binding protein that regulates tumor biological processes mainly by binding directly to RNA. To identify downstream target genes regulated by PTBP1, RNA molecules that can bind to PTBP1 proteins were captured by RIP and sequenced, yielding 4410 genes (Appendix A). Five candidate genes (*PGK1*, *YWHAH*, *RELB*, *MDM2*, *CSNK1G1*, and *SLC2A1*) have been identified by analyzing the intersection of 35 potential genes with the RIP-Seq data (Figure 3C). By inhibiting or overexpressing PTBP1, we evaluated the expression of five potential target genes in AGS cells. The results showed that inhibiting PTBP1 significantly downregulated PGK1, whereas overexpressing it upregulated PGK1. Nevertheless, there was no consistent effect on the expression of other candidate mRNAs (Figure 3D,E). Furthermore, GEPIA database analysis revealed a favorable association between PGK1 and PTBP1 expression in GC (Figure 3F). Additionally, PTBP1 has a positive regulatory effect on the protein level of PGK1 in both AGS and HGC27 cells (Figure 3G). The data thus suggest that PTBP1 increases the expression of PGK1. 

### 3.4. PTBP1 Enhances PGK1 mRNA Stability via RRM1 and RRM3 Domains

The molecular mechanism underlying the PTBP1 regulation of PGK1 was then explored further. First, RIP assay and RT-qPCR analysis confirmed that PTBP1 protein was specifically bound to endogenous *PGK1* mRNA in HGC27 cells (Figure 4A,B). Following this, we conducted an mRNA decay assay of PGK1. The results showed that the inhibition of PTBP1 significantly reduced the stability of *PGK1* mRNA in AGS and HGC27 cells (Figure 4C,D). PTBP1 is composed of 531 amino acids and possesses four RNA recognition motif (RRM) structural domains implicated in RNA binding. To ascertain the binding site of PTBP1 to *PGK1* mRNA, we generated four truncated mutants, all marked with a flag (Figure 4E). The efficiency of transfection and immunoprecipitation of the five PTBP1 mutant expression vectors was verified via Western blot (Figure 4F,G). The RIP experiments showed that the RRM1 and RRM3 domains were probably dominantly required for PTBP1-PGK1 interaction (Figure 4H). Taken together, these results suggest that the PTBP1 enhances *PGK1* mRNA stability via RRM1 and RRM3 domains.

### 3.5. PGK1 Promotes the Migratory Abilities of GC In Vitro

Next, we tried to investigate the effect of PGK1 on the migration in GC cells. First, we confirmed that PGK1 was upregulated in GC tissues and cell lines (SGC-7901, AGS, MKN74, and HGC27) compared to corresponding noncancerous tissues and gastric mucosal epithelial cell line (GES-1) (Figure 5A–C). Moreover, Kaplan–Meier survival analysis showed that the higher expression of *PGK1* mRNA was associated with shorter overall survival (OS) and time to first progression (FP) in GC patients (Figure 5D,E). This suggests that GC patients with higher PGK1 expression have a worse prognosis than those with lower expression. To investigate the biological function of PGK1 in GC, we silenced PGK1 using siRNA (si-PGK1) in AGS and HGC27 cells (Figure 5F,G). Transwell assay showed that the silencing of PGK1 expression resulted in a decrease in the migratory ability of GC cells (Figure 5H). These findings suggest that PGK1 is upregulated in GC and promotes the migratory ability of GC cells.

### 3.6. PTBP1 Promotes GC Cell Migration in a PGK1-Dependent Manner

We performed rescue assays to verify whether PTBP1 promotes gastric cancer cell migration in a PGK1-dependent manner. PTBP1-overexpressed AGS and HGC27 cells were transfected with siRNAs targeting PGK1 or negative control (si-NC). Western blot analysis confirmed that PGK1 knockdown significantly rescued the upregulation in PGK1 expression levels induced by PTBP1 overexpression (Figure 6A). The functional rescue assays demonstrated that the inhibition of PGK1 considerably hindered GC migration induced by PTBP1 overexpression (Figure 6B). This suggested that PGK1 played an important role in the promotion of gastric cancer cell migration by PTBP1. The activation of the EMT pathway is important for the promotion of tumor cell migration. Yang Hui et al. found that PGK1 could regulate SNAIL protein by interacting with HIF-2α, thereby activating EMT and promoting cell metastasis [10]. Our group has previously demonstrated that SNAIL plays an important role in activating the EMT pathway to promote the metastasis of gastric cancer cells [11]. Therefore, we further investigated whether PGK1 in gastric cancer also activates EMT and promotes metastasis of gastric cancer cells by regulating SNAIL expression. First, we verified that SNAIL expression was decreased after disturbing PGK1 in AGS and HGC27 GC cells (Figure 6C). We further examined SNAIL protein levels after the interference or overexpression of PTBP1. Western blot results showed that PTBP1 positively regulated the protein level of SNAIL (Figure 6D,E). We further investigated whether PTBP1 regulates SNAIL by regulating PGK1. The results showed that the disruption of PGK1 significantly rescued the increased SNAIL expression induced by PTBP1 overexpression (Figure 6F). Taken together, PTBP1 promotes GC cell migration in a PGK1-dependent manner.

## 4. Discussion

Recently, there has been accumulating evidence reinforcing the perception that the dysregulation or dysfunction of RBPs can lead to various human diseases, including cancers [12,13,14]. RBPs exert influence over a wide range of cancer-associated cellular phenotypes, including proliferation [15,16], apoptosis [17,18], senescence [19], metastasis [11,20], and angiogenesis [21], thereby contributing to tumor initiation, progression, and clinical prognosis [4]. PTBP1, an RNA-binding protein, belongs to the subfamily of ubiquitously expressed heterogeneous nuclear ribonucleoproteins (hnRNPs). The *PTBP1* gene is located on chromosome 19p13.3 in humans [6]. PTBP1 has been implicated in cancer development and progress across various tumor types. For instance, Aoran Luo demonstrated that PTBP1 interacted with lncRNA SFTA1P to regulate the degradation of *TPM4* mRNA, thereby promoting the proliferation, migration, and invasion of cervical cancer cells [22]. Haiyi Gong found that the knockout of the expression of PTBP1 could reduce cell proliferation, migration, and invasion, and significantly affect the cell cycle in osteosarcoma tumors [23]. Additionally, previous studies have reported the promotion of cancer metastasis by PTBP1 in colorectal cancer [24], hepatocellular carcinoma [25], and breast cancer [26]. In gastric cancer, Taro Sugiyama et al. found that PTBP1 promotes tumor progression by facilitating the conversion of *PKM* isoforms from *PKM1* to *PKM2*, leading to the Warburg effect in GC cells. This revealed an important function of PTBP1 in GC, for cancer-specific metabolism [27,28]. However, the effect of PTBP1 on the metastatic function in GC and the molecular mechanisms are unclear. In our study, we confirmed that PTBP1 was significantly highly expressed in GC and promoted cell migration in vivo and in vitro. This suggested that the high expression of PTBP1 in GC may be the key to promoting metastasis. However, the upstream regulatory mechanism of PTBP1 in GC has not been clarified in this study, which needs further exploration.

We further investigated the downstream target genes of PTBP1 in promoting metastasis of GC cells. In our study, *PGK1* was found to be the downstream target gene of PTBP1. Phosphoglycerate kinase (PGK) is a crucial enzyme that catalyzes ATP formation in the aerobic glycolysis pathway. While normally differentiated cells rely on mitochondrial oxidative phosphorylation for cellular energy supply, most tumor cells rely on aerobic glycolysis, a phenomenon known as the “Warburg effect”. PGK1 is a key enzyme in aerobic glycolysis and has an important function in the Warburg effect. PGK1 is expressed in all organisms and exhibits significant sequence conservation throughout evolution. Recent studies have found that PGK1 is intimately related to the development and progression of a variety of tumors by regulating glucose metabolism. For example, Zhong Chu demonstrated that PGK served as a crucial therapeutic target in breast cancer by mediating the Warburg effect [29]. PGK1 also can mediate glycolysis, particularly under hypoxic conditions, generating ATP for tumor cell proliferation [30]. The abnormal expression of PGK1 can affect the migration and invasion of tumor cells as well [31]. Siche Chen et al. found that the H19/miR-19a-3p/PGK1 pathway contributed to the regulation of aerobic glycolysis and cell proliferation in GC cells. This suggests that PGK1 is involved in aerobic glycolysis in GC. However, the upstream regulatory mechanisms and other functions of PGK1 in GC are unclear [32]. Our study found that PTBP1 stabilized *PGK1* mRNA and enhanced the expression of PGK1 and SNAIL, thereby promoting GC metastasis. In addition, given that both PTBP1 and PGK1 play important roles in glycolysis, it is suggested that PTBP1 may affect GC glycolysis by regulating PGK1. More research is needed to prove this.

PTBP1 is an RNA-binding protein that regulates tumor biological processes mainly by binding directly to RNA. Most RNA-binding proteins require at least one RNA-binding domain (RBD) to interact with RNAs [33]. These domains encompass the K homology domain (KH), the Zinc Finger type CCCH, the DEAD motif, the Arginine–Glycine–Glycine (RGG) box, and, most commonly, the RNA-recognition motif (RRM) [34]. PTBP1 is composed of 531 amino acids and consists of an N-terminal nuclear-shuttling domain and four RNA recognition motifs, which participate in the RNA-binding process [6]. In our study, the results suggest that PTBP1 increases *PGK1* mRNA stability by directly binding to *PGK1* mRNA and that the RRM1 and RRM3 domains may be responsible for the binding between PTBP1 and *PGK1* mRNA. Consistent with our results, studies by Ming Jen Wang also showed that RRM1 and RRM3 are essential components for the formation of complex I, thereby increasing the stability of *HIF 1α* mRNA [35]. This suggests that RRM1 and RRM3 of PTBP1 may play an important role in regulating mRNA stability. However, it has also been shown that RRM3 and RRM4 were the main regions of PTBP1 binding to RNA [36,37], whereas RRM1 and RRM2, as protein interaction domains, have almost no role in RNA binding [38]. This indicates the diversity and complexity of the binding between PTBP1 and RNAs. In our study, whether RRM1 and RRM3 of PTBP1 are indispensable for metastasis promotion needs further investigation. Taken together, these results suggest that the PTBP1/PGK1/SNAIL signaling axis plays a regulatory role in GC metastasis.

## 5. Conclusions

In summary, we verified that PTBP1 was upregulated in GC. Functionally, PTBP1 promotes the migration ability of GC cells in vitro and in vivo. Mechanistically, PTBP1 binds directly to *PGK1* mRNA, where RRM1 and RRM3 are the major RNA binding domains. PTBP1 promotes PGK1 expression by enhancing *PGK1* mRNA stability, thereby regulating the PGK1/SNAIL axis and promoting GC cell migration. In the future, PTBP1 may become a new target for the treatment of GC metastasis.

## Figures and Tables

**Figure 1 cells-13-00140-f001:**
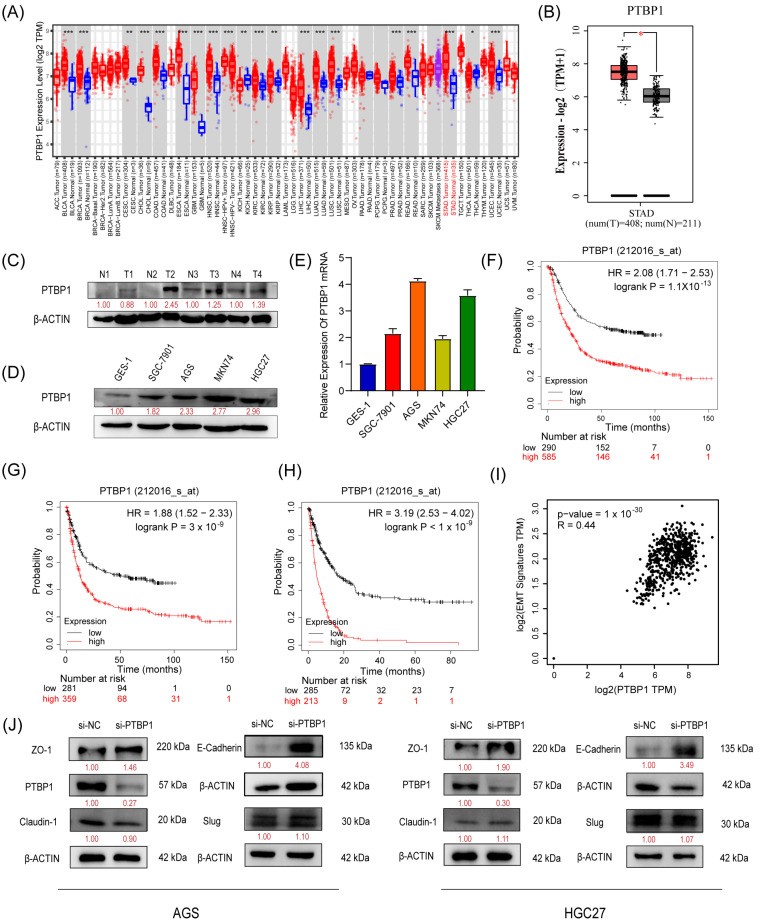
PTBP1 is highly expressed and is a poor prognostic factor in gastric cancer (**A**,**B**) Expression level of PTBP1 in cancers was analyzed using TIMER 2.0 (**A**) and GEPIA (**B**). The red asterisk indicated a significant difference between the two sets of data. (**C**) The protein levels of PTBP1 in GC tissues (*n* = 4) were measured via Western blot. (**D**,**E**) The protein levels and mRNA expression of PTBP1 in normal gastric mucosa epithelial cells and 4 gastric cancer cell lines. (**F**–**H**) Overall survival analysis (**F**), first progression survival analysis (**G**), and post-progression survival analysis (**H**) of GC patients by the Kaplan–Meier plotter online resource (http://kmplot.com/analysis/). The plot was generated according to the PTBP1 expression level. (**I**) The correlation analysis was conducted between PTBP1 and EMT-associated gene expression. (**J**) The protein levels of EMT-related genes (including ZO-1, E-Cadherin, Claudin-1, and Slug) were measured in AGS cells and HGC27 after interference with PTBP1. The uncropped blots are shown in Appendix A. All experiments were performed at least three times. *, *p* < 0.05; **, *p* < 0.01; ***, *p* < 0.001.

**Figure 2 cells-13-00140-f002:**
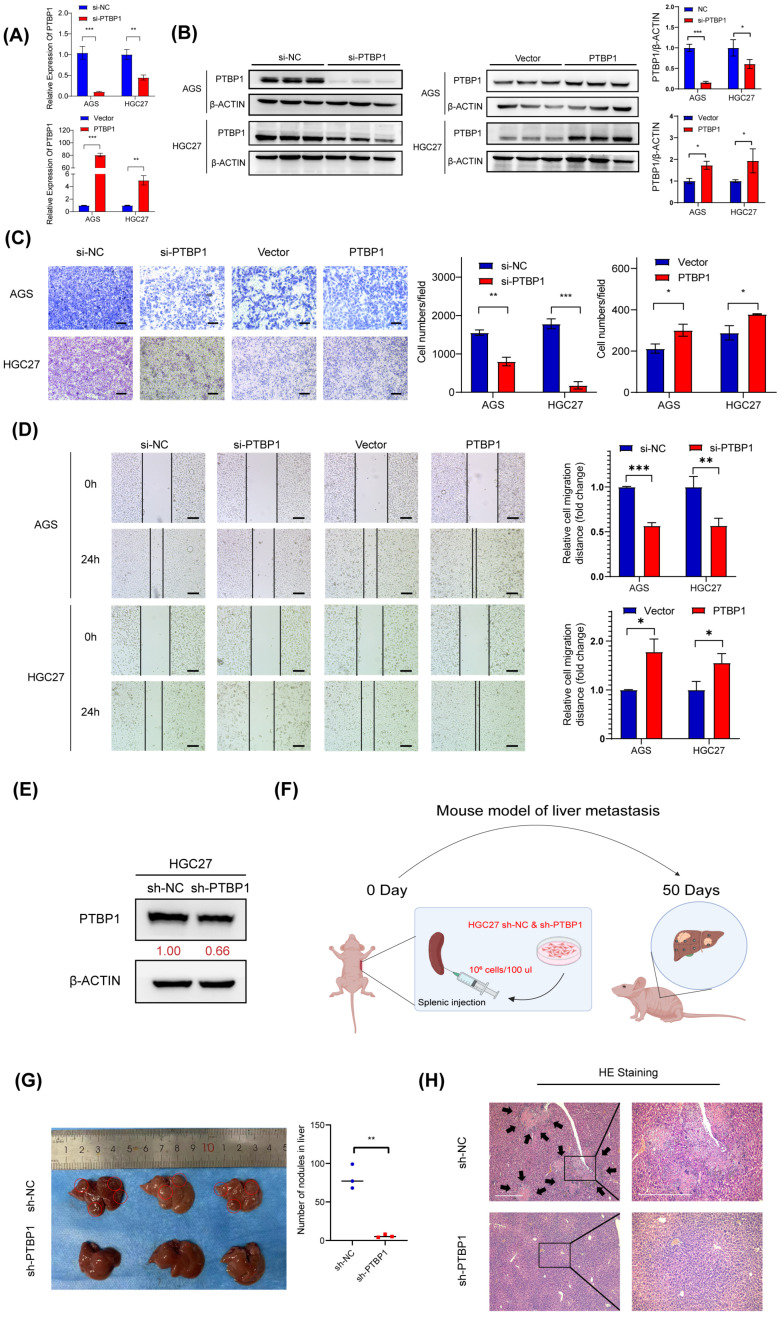
PTBP1 promotes GC metastasis in vitro and in vivo. (**A**,**B**) The mRNA expression and protein levels expression of PTBP1 mRNA in AGS and HGC27 cells treated with si-PTBP1 or PTBP1 overexpression plasmids. (**C**) Transwell assays were used to detect the migration abilities after knockdown or overexpression of PTBP1 in AGS and HGC27. Scale bar, 100 μm. (**D**) Wound-healing assays were used to detect the migration abilities after the knockdown or overexpression of PTBP1 in AGS and HGC27. (**E**) Efficiency of lentivirus sh-PTBP1 transduction in HGC27 cells. (**F**) Construction of a nude mouse model of GC with liver metastasis in which HGC27 cells were injected into the spleens of nude mice. (**G**) Representative images of mice livers in the nude mouse liver metastasis model (**left**) and the quantification of liver metastatic colonization (**right**). (**H**) Representative images of pulmonary metastatic nodules after H&E staining in liver metastasis model. The black arrows were used to emphasize pulmonary metastases in the nude mouse liver metastasis model. Scale bar, 500 μm. The uncropped blots are shown in Appendix A. All experiments were performed at least three times. Bars represent means ± SD. *, *p* < 0.05; **, *p* < 0.01; ***, *p* < 0.001.

**Figure 3 cells-13-00140-f003:**
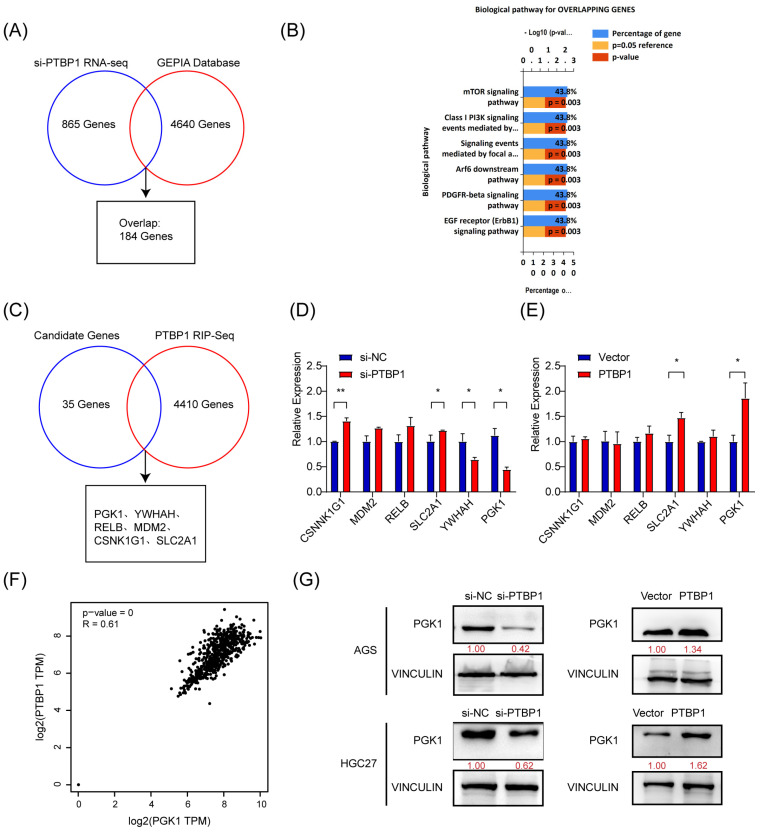
PTBP1 positively regulates PGK1 in gastric cancer. (**A**) Venn diagram showing the overlap between si-PTBP1 RNA-seq and GEPIA database. (**B**) Biological pathway analysis in FunRich was performed to screen potential biological pathways after the knockdown of PTBP1 in AGS cells. (**C**) Venn diagram showing the strategy to screen potential targets of PTBP1. (**D**,**E**) The mRNA expression of potential targets were measured in AGS cells with the knockdown or overexpression of PTBP1. (**F**) The correlation analysis was conducted between PTBP1 and PGK1. (**G**) The protein levels of potential targets were measured in AGS cells and HGC27 with the knockdown or overexpression of PTBP1. The uncropped blots are shown in Appendix A. All experiments were performed at least three times. Bars represent means ± SD. *, *p* < 0.05; **, *p* < 0.01.

**Figure 4 cells-13-00140-f004:**
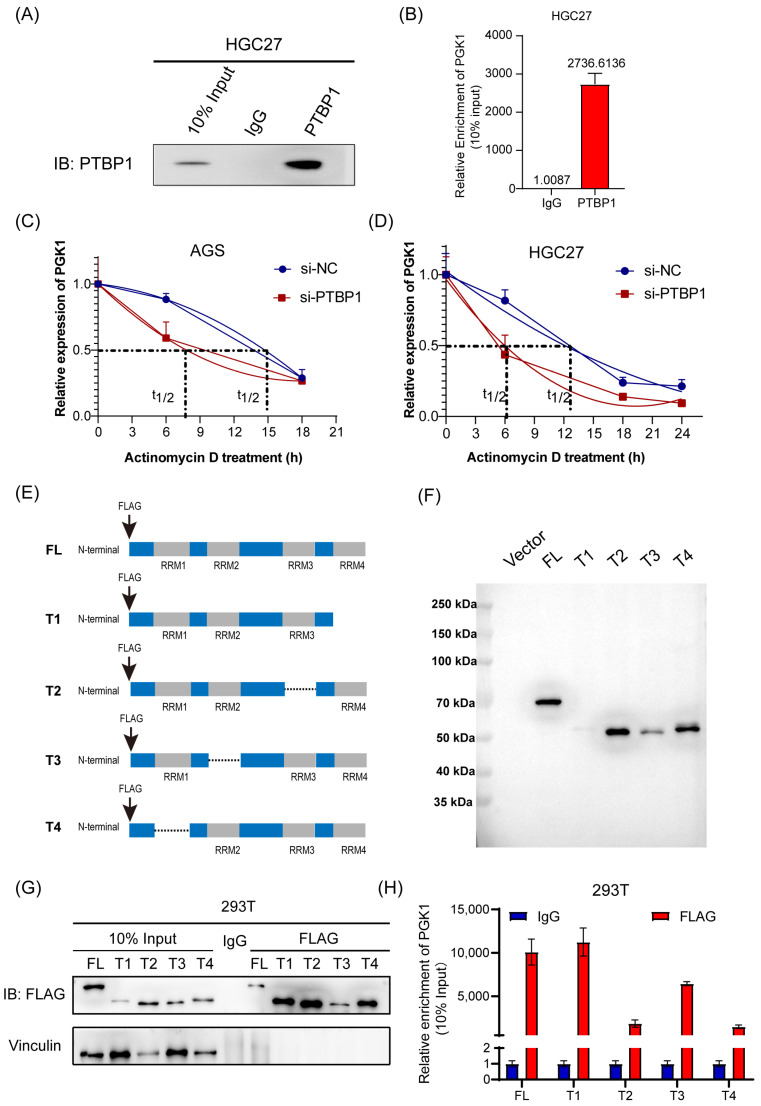
PTBP1 enhances *PGK1* mRNA stability via RRM1 and RRM3 domains. (**A**) PTBP1 protein was pulled down via RIP assay in HGC27 cells. (**B**) Enrichment of *PGK1* mRNA pulled down by PTBP1 protein was detected via RT-qPCR. (**C**,**D**) The actinomycin D assay showed the stability of *PGK1* mRNA in AGS (**C**) and HGC27 cells (**D**) with PTBP1 knockdown at the indicated time point. (**E**) Schematic diagram of PTBP1 truncated mutant construction. (**F**) Schematic diagrams of flag-tagged PTBP1 mutants were shown. (**G**) The full-length or truncated PTBP1 proteins pulled down by anti-flag were verified via Western blot. (**H**) RIP analysis of *PGK1* mRNA enrichment pulled down by flag in 293T cells transfected with full-length or truncated PTBP1 was verified via RT-qPCR. The uncropped blots are shown in Appendix A.

**Figure 5 cells-13-00140-f005:**
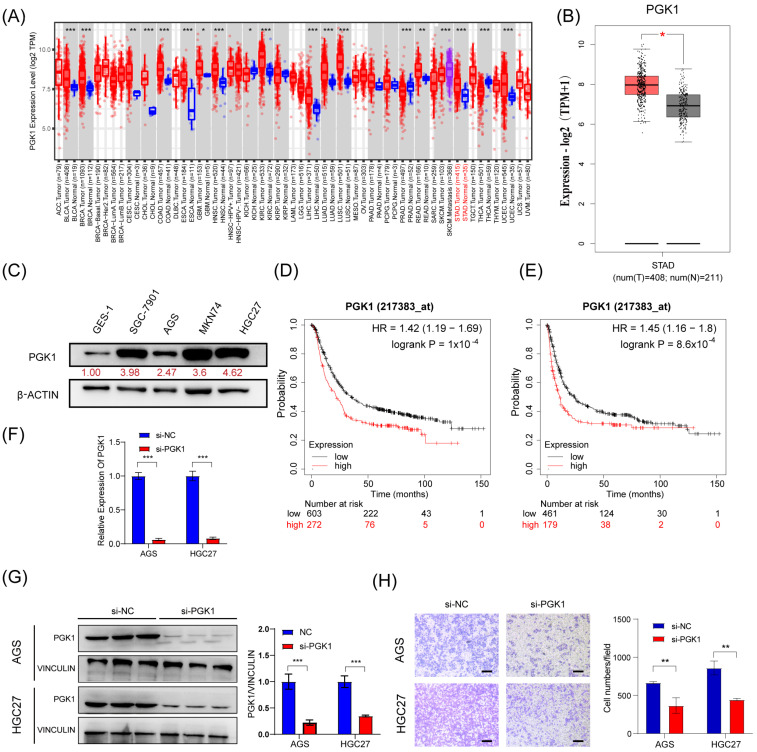
PGK1 promotes the migratory abilities of GC in vitro. (**A**,**B**) Expression level of PGK1 in cancers was analyzed using TIMER 2.0 (**A**) and GEPIA (**B**). The red asterisk indicated a significant difference between the two sets of data. (**C**) The protein levels of PGK1 in normal gastric mucosa epithelial cells and 4 gastric cancer cell lines were detected via Western blot. (**D**,**E**) Overall survival analysis (**D**) and first progression survival analysis (**E**) of GC patients by the Kaplan–Meier plotter online resource (http://kmplot.com/analysis/). The plot was generated according to the PGK1 expression level. (**F**,**G**) The expression of *PGK1* mRNA and protein level in AGS and HGC27 cells treated with PGK1 knockdown by siRNAs. (**H**) Transwell assays were used to detect the migration abilities after the knockdown of PGK1 in AGS and HGC27 cells. Scale bar, 100 μm. The uncropped blots are shown in Appendix A. All experiments were performed at least three times. Bars represent means ± SD. *, *p* < 0.05; **, *p* < 0.01; ***, *p*< 0.001.

**Figure 6 cells-13-00140-f006:**
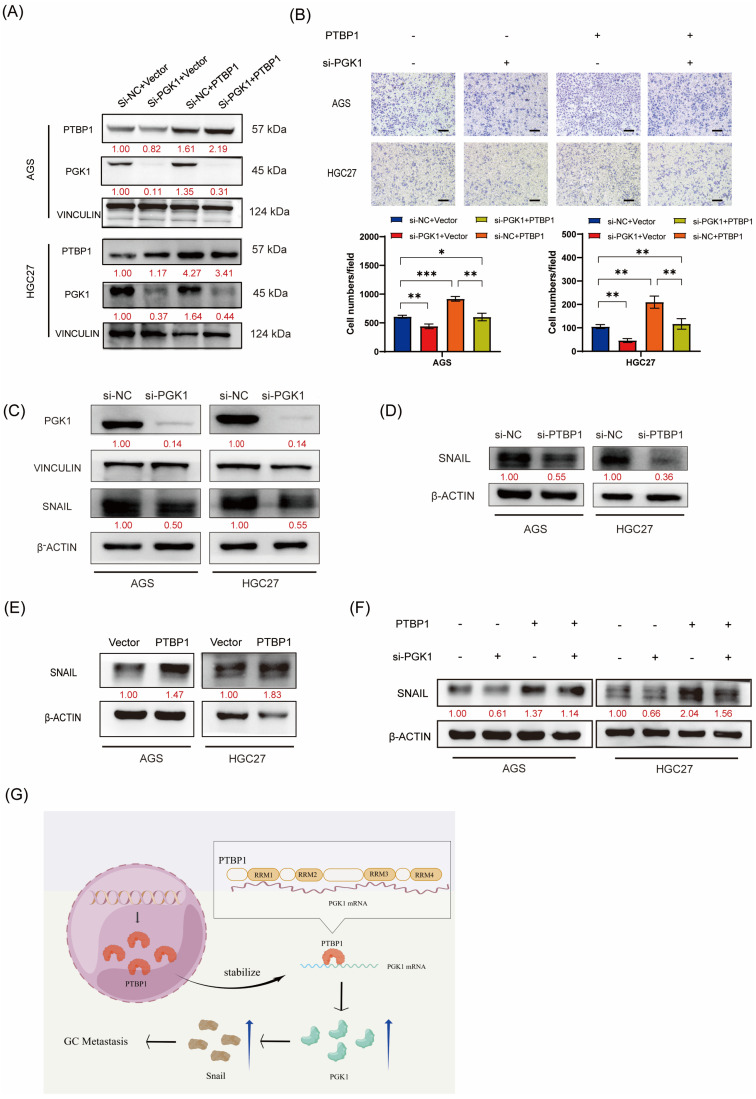
PTBP1 promotes GC cell migration in a PGK1-dependent manner. (**A**) The transfection efficiency of the rescue assay was verified via Western blotting in AGS and HGC27 cells. (**B**) Transwell assays were used to detect the migration abilities of AGS and HGC27 cells upon the overexpression of PTBP1 combined with PGK1 knockdown. Scale bar, 100 μm. (**C**) The protein level of SNAIL was detected via Western blotting in AGS and HGC27 cells after the knockdown of PGK1. (**D**,**E**) SNAIL protein levels were determined after the deletion or overexpression of PTBP1. (**F**) Western blot assay showed the protein levels of SNAIL in AGS and HGC27 cells upon the overexpression of PTBP1 combined with the PGK1 knockdown. (**G**) This schematic illustrates the mechanism of action of PTBP1 in gastric cancer (GC) to promote metastasis. The uncropped blots are shown in Appendix A. All experiments were performed at least three times. Bars represent means ± SD. *, *p* < 0.05; **, *p* < 0.01; ***, *p* < 0.001.

## Data Availability

The datasets used and analyzed in this paper are available from the corresponding author upon reasonable request.

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
