# Peer review of "RNA Binding Protein PTBP1 Promotes the Metastasis of Gastric Cancer by Stabilizing PGK1 mRNA"

_cells, 2024, doi:10.3390/cells13020140_

Round 1

Reviewer 1 Report

Comments and Suggestions for Authors

This study was planned to demonstrated that PTBP1 was significantly highly expressed in GC and was strongly associated with poor prognosis of GC patients.

1. Why did you initially pick PTBP1 as a candidate?

2. Usually, there are various methods of gastric cancer metastasis models, such as subcutaneous injection, when injected intraperitoneally. The transition model injected into Spleen is unfamiliar. What reference does it have?

Comments on the Quality of English Language

good job.

Reviewer 2 Report

Comments and Suggestions for Authors

This manuscript investigates the role of Polypyrimidine Tract Binding Protein 1 (PTBP1) in gastric cancer, highlighting its overexpression and correlation with poor prognosis. The study intriguingly links higher levels of PTBP1 with increased stability of Phosphoglycerate Kinase 1 (PGK1), a downstream target of PTBP1. Furthermore, it elucidates the role of PGK1 in regulating the SNAIL protein, thereby activating the epithelial-mesenchymal transition (EMT) pathway, which is crucial for gastric cancer metastasis.

While the topic is of significant interest and relevance, the manuscript requires several revisions for clarity and depth before it is ready for publication:

Major comments

1: It is essential to provide quantitative values alongside the Western blot results to enhance the clarity and scientific rigor of the findings. The current presentation of internal control bands shows noticeable heterogeneity, leading to ambiguity in the results. Although the text mentions changes in protein expression as evidenced by Western blotting, this is not clearly reflected in the corresponding figures (e.g., Fig.2E, 3G, 5C, and Fig.6). To resolve this discrepancy, I recommend including densitometric analysis of the bands to provide a quantifiable measure of the changes in protein expression. This approach will ensure a more accurate and reliable interpretation of the data.

2: Figure 2H fails to demonstrate the expected decrease in tumor size. The author should conduct tumor size measurements and provide a detailed pathological explanation to support the observed results.

3:While the authors suggest that PTBP1 is linked to the activation of Epithelial-Mesenchymal Transition (EMT), their evidence is limited to variations in SNAIL expression. It is recommended that they also examine the expression of other EMT markers, particularly E-cadherin, to comprehensively substantiate their claim.

4: In previous studies, a relationship between gastric cancer and PTBP1 (PMID: 33070451 and 27696637), particularly in the context of microRNAs and cancer-specific metabolism, has been suggested. It would be beneficial to discuss these findings in relation to our current research.

Minor comments

1: Page 11, line 306: Replace "PTBP1" with "PGK1". The sentence should read: "Transwell assay showed that silencing of PGK1 …"

2: In Figures 1F-H and 5D-E, the author needs to clearly specify what each graph represents, whether it is Overall Survival (OS), Progression-Free Survival (PFS), or Post-Progression Survival (PPS).

3: The arrangement of samples in Figure 6B should be consistent with the order presented in the other figures.

4: Page 1, line 42: The abbreviation for 'RNA binding proteins' should be corrected to 'RBPs'. The sentence should read: "More than 2000 RNA binding proteins (RBPs) …

5: Page 4, lines 167-169: The text should be in the present tense. It should read: "After this step, the synthesis of second-strand cDNA occurs … Subsequently, the cDNA fragment undergoes …" Please have the document proofread by a native English speaker once.

6: Page 10, line 285: There is a typographical error with 'PTBP1' being duplicated.

7: Figure 3F is not described in the figure legend. The legend should be updated to include an explanation of Figure 3F.

8: The author should provide the approval numbers for clinical and animal experiments. This information is essential for verifying the ethical compliance of the study.

Comments on the Quality of English Language

Please have the document proofread by a native English speaker once.

Round 2

Reviewer 2 Report

Comments and Suggestions for Authors

I believe that the revisions have generally improved the document. However, a few points require further consideration, so please check them.

1. This study focuses on the relationship between PTBP1 and PGK1. Please show how PTBP1 and PGK1 expressions are correlated in gastric cancer cell lines. Although PTBP1 is highly expressed in HGC27, as seen in Figure 1D, it seems similar to GES-1 in Figure 5C. Also, check the mRNA expression of PGK1 if needed. I understand that not all cell lines may show expected results, but this is a topic worth discussing.

Comments on the Quality of English Language

Some periods are duplicated in lines 228 and 230, page 7. I recommend more English editing.

Round 3

Reviewer 2 Report

Comments and Suggestions for Authors

The author’s revision revealed the correlation between PTBP1 and PGK1.